# Comparative Analysis of Osteointegration in Hydroxyapatite and Hydroxyapatite-Titanium Implants: An In Vivo Rabbit Model Study

**DOI:** 10.3390/jfb15070181

**Published:** 2024-06-29

**Authors:** Renata Maria Văruț, Luciana Teodora Rotaru, Flavius Nicușor Truicu, Cristina Elena Singer, Iliescu Iulian-Nicolae, Alin Iulian Silviu Popescu, Cristina Popescu, Cristina Teisanu, Gabriela Sima, Oana Elena Nicolaescu

**Affiliations:** 1Research Methodology Department, Faculty of Pharmacy, University of Medicine and Pharmacy of Craiova, 200349 Craiova, Romania; renata.varut@umfcv.ro; 2Emergency Medicine and First Aid Department, Faculty of Medicine, University of Medicine and Pharmacy of Craiova, 200349 Craiova, Romania; luciana.rotaru@umfcv.ro (L.T.R.); flavius.truicu@rmu.smurd.ro (F.N.T.); 3Department of Mother and Baby, University of Medicine and Pharmacy of Craiova, 200349 Craiova, Romania; singercristina@gmail.com (C.E.S.); iliescuiuliann@gmail.com (I.I.-N.); 4Department of Internal Medicine, University of Medicine and Pharmacy of Craiova, 200349 Craiova, Romania; 5Discipline of Anatomy, Department of Anatomy, University of Medicine and Pharmacy of Craiova, 200349 Craiova, Romania; 6Department of Materials Science, Faculty of Mechanics, University of Craiova, Calea Bucuresti 107, 200512 Craiova, Romania; cristina.teisanu@edu.ucv.ro (C.T.); gabriela.sima@edu.ucv.ro (G.S.); 7Department of Pharmaceutical Technique, University of Medicine and Pharmacy of Craiova, 200349 Craiova, Romania; oana.nicolaescu@umfcv.ro

**Keywords:** bioceramic, two-steps sintering, osteointegration

## Abstract

The study evaluates the osteointegration of hydroxyapatite (HAp) and hydroxyapatite-titanium (HApTi) biocomposites implanted in the femurs of rabbits. The biocomposites were fabricated using powder metallurgy and subjected to a two-step sintering process. Scanning electron microscopy (SEM) was employed to analyze the morphology, while mesenchymal stem cells were cultured to assess cytotoxicity and proliferation. In vivo experiments involved the implantation of HAp in the left femur and HApTi in the right femur of twenty New Zealand white rabbits. Computed tomography (CT) scans, histological, immunohistochemical, and histomorphometric analyses were performed to assess bone density and osteoblast activity. Results demonstrated that HApTi implants showed superior osteointegration, with higher peri-implant bone density and increased osteoblast count compared to HAp implants. This study concluded that HApTi biocomposites have potential for enhanced bone healing and stability in orthopedic applications.

## 1. Introduction

In recent times, there has been a growing interest in the development of biomimetic materials aimed at enhancing cellular adhesion and reducing the integration time of bone implants. These materials are designed to closely mimic the natural extracellular matrix, providing an optimal environment for cell activities such as adhesion, proliferation, and differentiation. Their biocompatibility is crucial to avoid immune responses that can lead to inflammation, proteolytic enzyme production, and granuloma formation around the implant, ultimately causing bone resorption and osteolysis [1,2].

Hydroxyapatite (HAp) is a biomaterial renowned for its excellent biocompatibility and bioactivity, making it a popular choice in the production of bone grafts and the coating of metallic orthopedic components. Its chemical composition closely resembles that of human bone mineral, which not only supports the adhesion and growth of osteoblasts but also facilitates the integration of implants with the surrounding bone tissue [3]. After implantation, HAp releases calcium and phosphate ions that facilitate the adherence of the implant to the surrounding tissue by promoting the formation of a functional connective structure. These ions support the deposition of new bone matrix and the growth of osteoblasts, the cells responsible for bone formation [4]. There are significant disadvantages associated with pure HAp, primarily its weak mechanical properties and the difficulty of producing hard ceramic biocomposites. To address these issues, reinforcement with particles, short fibers, and long fibers has been employed to create HAp biocomposites with enhanced mechanical properties. One effective approach is the incorporation of bio-inert metal particles, such as titanium (Ti), which significantly improves both the mechanical strength and the biological performance of HAp composites. Titanium particles enhance the composite’s durability while maintaining its biocompatibility, making it a robust material for orthopedic and dental applications. A major challenge in the field of implantology and bone reconstruction is the identification of biocomposites that combine the advantages of their constituent components. Porous HAp exhibits a unique structure that significantly enhances osteointegration compared to its non-porous counterpart. The presence of pores in HAp creates a conducive environment for bone ingrowth, vascularization, and nutrient diffusion, all of which are essential for effective bone healing and integration. The porous structure allows for the migration and proliferation of osteoblasts, the cells responsible for bone formation, facilitating a stronger and more stable bond between the implant and the bone. This characteristic is especially important in orthopedic and dental applications, where the long-term success of the implant is dependent on its ability to integrate seamlessly with the host bone [5]. Non-porous HAp, while still biocompatible and osteoconductive, lacks the surface architecture necessary to promote substantial bone ingrowth. The solid structure of non-porous HAp limits the space available for osteoblast activity and restricts vascularization, which is critical for providing the necessary nutrients and oxygen to the bone-forming cells. As a result, implants made from non-porous HAp may exhibit weaker integration and a higher risk of implant failure due to insufficient bonding with the surrounding bone [6]. The importance of porosity in HAp for bone integration of implants lies in several key factors. Porous HAp provides a scaffold-like structure that supports the attachment and growth of osteoblasts. The interconnected pores create pathways for cell migration, allowing osteoblasts to colonize the implant surface and initiate bone formation. This enhanced osteoconduction is one of the primary reasons porous HAp is superior in promoting osteointegration. Additionally, the porous nature of HAp facilitates the ingrowth of blood vessels, which is crucial for supplying nutrients and oxygen to the newly formed bone tissue. Improved vascularization also aids in the removal of metabolic waste, creating a healthier environment for bone regeneration. Without adequate vascularization, the integration process can be significantly impaired, leading to potential implant failure. The increased surface area of porous HAp offers more binding sites for bone matrix proteins and cells. This higher surface area enhances the mechanical interlocking between the bone and the implant, resulting in a more stable and durable integration. This is in contrast to non-porous HAp, which provides limited surface area and thus fewer opportunities for strong bone bonding. Furthermore, porous HAp can better mimic the mechanical properties of natural bone, providing a more uniform stress distribution at the implant-bone interface. This reduces the risk of stress shielding, where the implant takes on too much load, leading to bone resorption and weakening. By distributing stress more evenly, porous HAp helps maintain the integrity and strength of the surrounding bone. The interconnected pores in HAp also facilitate the diffusion of essential nutrients and growth factors, promoting a conducive environment for bone healing and regeneration. This is particularly important in the early stages of implantation when the bone tissue is still forming and integrating with the implant. Effective nutrient diffusion ensures that the osteoblasts and other cells involved in bone formation receive the necessary sustenance to perform their functions optimally [7].

The objective of this research was to evaluate osteointegration following femoral implantation in rabbits of two types of biocomposites: HAp and titanium-hydroxyapatite (HApTi). Although there are significant differences in the size and shape of human and rabbit bones, we chose to conduct experiments on rabbits due to their availability, as well as the lower costs of acquisition and pre- and post-operative care, in compliance with current standards.

## 2. Materials and Methods

### 2.1. The Method of Obtaining Biocomposites

The biocomposite samples were synthesized utilizing powder metallurgy techniques. HAp served as the matrix material, characterized by its chemical formula Ca_5_(PO_4_)_3_(OH), and was sourced from Merck, featuring an average particle size of 200 nm. To fabricate HApTi, the HAp powder was reinforced with titanium hydride particles (Merck; approximately 100 μm), maintaining a mixing ratio of 75% HAp and 25% titanium hydride mass percentages. Initially, the HAp was dried, followed by a calcination process conducted in air at 900 °C for 1 h. The calcined matrix and reinforcing powder particles were then blended in a planetary ball mill for 30 min, using stainless steel grinding balls (5 mm in diameter) at a ball-to-powder ratio of 2:1, with ethanol as the milling medium (1 mL/1 g powder mixture). The resulting mixture was dried at 200 °C for 1 h in a conventional oven, and the grinding balls were subsequently removed from the powder mixture. The formation of the biocomposite samples was achieved through cold uniaxial and unidirectional compaction, applying a compaction pressure of 120 MPa. This process was conducted in a metal mold with an inner diameter of 10 mm, using the A009 universal materials testing equipment (100 kN) (Galdabini S.p.A., Cardano al Campo, Italy), equipped with TCSoft2004Plus software version 2004. The compacted samples underwent a two-stage sintering heat treatment using a Nabertherm HTCT 08/16 laboratory furnace (Nabertherm GmbH, Lilienthal, Germany), with a maximum temperature of 900 °C (Figure 1).

A high-purity argon gas atmosphere (99.98% pure) was maintained throughout the treatment process. In the first stage of the two-step thermal sintering treatment, samples were rapidly heated to 900 °C for 1 min to initiate diffusion between HAp and titanium hydride, achieving intermediate density. In the second stage, the samples were cooled to 800 °C and held for 10 h, allowing final densification without increasing the HAp particle diameter, thereby consolidating the composite material while preserving its fine microstructure. The detailed processing technology is described by Gingu et al. [8] (Figure 2 and Figure 3).

### 2.2. Scanning Electron Microscopy (SEM)

SEM was used to examine the samples’ morphological characteristics, including particle dimensions, form, agglomeration tendency, and porosity. High-resolution micrographs were captured with an FEI Inspect F50 scanning electron microscope (FEI Company, Hillsboro, OR, USA) at 30 keV and various magnifications. As the samples were non-conductive, they underwent a metallization process, coating them with a 4 nm thick gold film for 60 s to achieve electrical conductivity. The samples were mounted on an aluminum holder using conductive carbon tape, and the analysis was conducted under vacuum, measuring the energy of secondary electrons generated by the primary electron beam’s interaction with the sample surfaces.

### 2.3. X-rays Diffraction

The sintered parts were microscopically analyzed by SEM, FEI Co. (Hillsboro, OR, USA), model Quanta Inspect S with EDS facility. The structural composition of the samples was determined by X-ray diffraction (XRD) using a Bruker D8 advance diffractometer (Bruker Corporation, Billerica, MA, USA.) with Kα (λ = 1.5418 Å) radiation.

### 2.4. In Vitro Study

For the analysis of cytotoxicity and proliferation effects, biomaterials were cultured with mesenchymal stem cells (MSCs). The biomaterials were seeded with MSCs for 48 h and fixed using 2.5% glutaraldehyde for one hour. Post-fixation, the samples were thoroughly washed with phosphate-buffered saline (PBS), dehydrated through treatment with varying concentrations of ethanol, and dried in a vacuum. Before fixation, photographs were taken around the biomaterials, using a cover slip as a control. The morphology of the MSCs was observed using SEM, FEI Inspect F50 (FEI Company, Hillsboro, OR, USA).

### 2.5. Biocompatibility Evaluation toward Cell Culture

#### 2.5.1. Sterilization of Materials

Materials underwent sterilization through incubation in a 1% penicillin-streptomycin solution in PBS for 2 h at room temperature. Post-sterilization, these materials were used for cell seeding and subsequent experimental procedures.

#### 2.5.2. Culture of Mesenchymal Stem Cells (MSCs)

MSCs were extracted from bone marrow samples collected from patients having prosthetic surgery at the Emergency Hospital in Craiova, Romania, with informed consent and ethical approval. The study received approval from the Hospital’s Ethics Committee (Approval No. 68/2016). The bone marrow was processed using Ficoll to isolate MSCs, which were then cultured in low-glucose DMEM with supplements. The culture medium was replaced every 3 days, and cells were expanded for about 10 days at 37 °C and 5% CO_2_, maintaining confluence below 80%. At passage two, cells were validated by flow cytometry, then used in experiments or cryopreserved. For experiments, cells were seeded at a density of 5000 cells/cm^2^.

#### 2.5.3. Immunofluorescence Microscopy

The samples were analyzed for adhesion and proliferation markers using fluorescence microscopy. Five days post-seeding, samples were washed with PBS, fixed with 4% paraformaldehyde for 20 min, permeabilized with 0.2% Triton-X-100 (Sigma-Aldrich, St. Louis, MO, USA), and blocked with 0.5% BSA. They were then incubated with the primary antibody anti-Ki67 for 30 min, followed by secondary antibodies conjugated to Alexa Fluor 594 (Thermo Fisher Scientific, Waltham, MA, USA) for vinculin, and stained with Alexa Fluor 488 (Thermo Fisher Scientific, Waltham, MA, USA) Phalloidin for actin filaments. After washing and staining with Hoechst, the samples were mounted on slides with Prolong Antifade solution (Invitrogen, Carlsbad, CA, USA) and affixed with Loctite glue for automatic scanning with the TissueFAXS iPlus system (TissueGnostics GmbH, Vienna, Austria). Further analysis was performed using a Zeiss Axio Imager Z1 microscope (Boston Microscopes, Wilmington, MA, USA) to highlight focal contacts.

#### 2.5.4. Image Cytometric Analysis

The TissueFAXSiPlus technology platform from TissueGnostics (Vienna, Austria) enables the scanning and reconstruction of specimens mounted on microscope slides utilizing the TissueFAXS Slides module and PCO PixelFly camera under standardized conditions. Quantitative protein expression analysis was performed with the TissueQuest software, (version 6.0), which detects cell nuclei stained with DAPI. Parameters for image segmentation included average nuclear size, discrimination area, pixel gray level, and background threshold. Each fluorescence channel (FITC, TxRed, DAPI) was measured separately, with regions of interest (ROIs) defined to exclude the sample edges. General settings, including gating strategy and cutoffs for positive and negative cells, were configured using a representative image. The detailed protocol is described by Sima et al. [10].

### 2.6. In Vivo Study

#### 2.6.1. Animals Used, Anesthesia, and Operative Technique

For the experiment, male New Zealand white rabbits, aged 6 months and weighing between 3000 and 3500 g, were utilized. Throughout the experimental period, the rabbits were housed individually in plastic cages maintained at 25 °C with a 12 h light/dark cycle, and were provided with a normal diet and water ad libitum.

The protocol for implant placement, postoperative care, and euthanasia for harvesting bone tissues had been successfully employed in previous research projects and was approved by the Ethics Committee of UMF Craiova (No. 134/2019). For the procedure, the rabbits were sedated with subcutaneous injections of fentanyl (0.1 mL/kg) and midazolam (2 mg/kg) (Figure 4). Anesthesia was maintained during surgery with fentanyl diluted in saline solution (1 mL fentanyl to 9 mL saline), and supplemented with 1% xyline (5 mL) administered at the incision site (Figure 5). Prior to surgery, the incision site was shaved, cleaned with soap, water, and betadine solution, and then covered with a sterile field (Figure 6). A 5 cm incision was made on the anterior surface of the proximal femur, extending through the epidermis, dermis, and fascial layers to expose the periosteum-covered femur (Figure 7). The periosteum was incised and removed using a rasp (Figure 8). A cavity was created in each femur, extending through the cortical bone to the medullary canal, using a Stryker Core Reamer orthopedic motor at low speed (Figure 9 and Figure 10) [11].

#### 2.6.2. Experimental Design

Twenty rabbits were used in the in vivo study. For each animal, excavations were made in each femur, into which implants (3 mm in diameter, 5 mm in length) were inserted.

HAp was implanted in the left femur, while HApTi was implanted in the right femur; the implants in the left femur served as controls for the implants in the right femur. The two samples were implanted in different femurs of the same animal to exclude interindividual differences. After the incisions were made, the structures were sutured using 4-0 Dexon suture material (Covidien, Dublin, Ireland) (Figure 11). Following the surgical procedure, two doses of buprenorphine diluted in saline were administered subcutaneously at a dosage of 0.05 mg/kg, with an interval of four hours between doses. The surgical wound was inspected and dressed daily until it fully healed [12].

#### 2.6.3. Computed Tomography of the Femurs

All CT scans were conducted using a Siemens CT scanner (Siemens Healthineers, Erlangen, Germany) with uniform settings for all rabbits: 130 kV, 90 mA, 0.5 mm section thickness, and 0.3 mm section increment. The integration time was 0.5 s with a total scan time of 2 min. The data were stored in JPEG format. To evaluate bone density, the Onis 2.3.5 software was used. A circular region of interest was employed to measure bone density on the CT sections. The location and size of the region of interest were standardized for each measurement performed, and the measured values were expressed in Hounsfield units.

#### 2.6.4. Histological and Immunohistochemical Analysis

Histological analysis was performed to evaluate the degree of osteointegration of the composites, osteoformation, and bone tissue biocompatibility. Eight weeks post-implantation, the animals were euthanized, and the femoral bones were processed in accordance with the established protocol, producing slides stained with hematoxylin–eosin, which were analyzed under an optical microscope. To prepare the slide for examination, several steps were followed, including the paraffin embedding technique and the histological staining technique with hematoxylin–eosin. The paraffin embedding technique involved numerous steps for the biological samples. Decalcification was achieved by immersing the femur with the implant in a buffered 10% ethylenediaminetetraacetic acid solution (pH 7.4) for two months. Dehydration of the anatomical pieces was carried out by placing them in containers with ethanol at progressively higher concentrations. In tightly sealed containers, the anatomical pieces were immersed twice in 75° alcohol, twice in 90° alcohol, and twice in absolute ethyl alcohol. The clarification of the anatomical pieces was carried out with xylene, which served to remove traces of alcohol from the tissues. Paraffin embedding was performed using melted paraffin, which penetrated the tissues, providing them with a homogeneous consistency, essential for making thin sections. Actual embedding into paraffin blocks was performed using plastic molds, where the pieces were encased in melted paraffin. Sectioning of the blocks was performed using a microtome, with sections having a thickness of 4 μm. The attachment of sections to slides and their drying were carried out by placing the sections on the surface of well-cleaned and degreased slides. Staining with a mixture of hematoxylin–eosin allowed for tissue recognition, due to the different coloring of their components: cell nuclei appeared intensely colored in blue-violet, cytoplasm appeared in light violet, collagen fibers had a pale pink color, and elastic and reticulin fibers were not highlighted [13].

For the immunohistochemical analysis, primary antibodies and anti-OC antibody [OCG3] (ab13420) were used as follows. To conduct the immunohistochemical study, histological sections were collected on histological slides coated with poly-L-lysine (Sigma Aldrich) to increase the adhesion of the sections to the slides, after which they were transferred to an incubator at 45 °C and kept overnight (12 h). The next day, the classic immunohistochemical protocol was employed, which involved initially deparaffinizing the sections by immersion in xylene (3 × 5 min). After deparaffinization, the sections were hydrated with 100%, 90%, and 70% alcohol (3 × 5 min), then introduced into distilled water for 10 min. This was followed by antigen retrieval via immersion in proteinase K for 20 min at a controlled temperature (37 °C). The samples were then cooled and washed with distilled water (3 × 5 min), and then with 3% hydrogen peroxide (6 mL H202 + 200 mL distilled water) for 10 min to block endogenous peroxidase. This was followed by washing with distilled water (2 × 5 min) and washing with SSTF (5 min). Blocking of non-specific sites was achieved by immersion in milk for 30 min (6 g Nan milk + 200 mL SSTF, mixed using a shaker). Then, the sections were incubated with the primary antibody for 18 h (overnight) in a refrigerator at 4 °C. The primary antibody was diluted in SSTF at a ratio of 1:50. The next day, the sections were kept at room temperature for 30 min, washed with SSTF (3 × 5 min), and the secondary antibody was applied (ImmPRESS™ HRP Anti-Mouse IgG (peroxidase enzyme) Polymer Detection Kit, made in Goat, MP-7452-15, Vector Lab (Burlingame, CA, USA)) for 60 min at room temperature. This was followed by washing with SSTF (3 × 5 min), development with diaminobenzidine (Histofine^®^ 3,3′-diaminobenzidine-3S kit, 415192F (Nichirei Biosciences Inc., Tokyo, Japan)), washing with distilled water (2 × 5 min), and staining with hematoxylin for contrast for 12 min. Afterwards, the dehydration was carried out in 70%, 90%, and 100% alcohol for 5 min each, followed by clarification in xylene for 30 min and mounting of the slides using BioMount mounting medium [14,15].

#### 2.6.5. Histomorphometric Analysis

Histomorphometric analysis was conducted on immunohistochemically stained OC sections using the ImagePro Plus 6 software (Media Cybernetics, Inc., Silver Spring, MD, USA). This involved counting the osteoblasts surrounding the area corresponding to the 5 mm of the implant. Additionally, a region of interest was established encompassing a distance of 1 cm of cortical bone adjacent to the implant [16,17].

#### 2.6.6. Statistical Analysis

Experimental measurements were conducted on three specimens, with results expressed as the mean of three analyses ± standard deviation. To assess possible differences between the number of osteoblasts per μm^2^ in the left and right femur, the paired Student’s *t*-test was applied. The significance of the comparisons was defined by *p*-values equal to or less than 0.05, t-values less than 2, and degrees of freedom greater than 200. This procedure was conducted using the *t*-test function in Microsoft Excel (version 2019) [18].

## 3. Results

### 3.1. Scanning Electron Microscopy of the Implant Surfaces

Figure 12A–D present SEM images of the surface of HAp samples, revealing various structural characteristics. In Figure 12A, visible microcracks can be observed, such as one extending from the upper left to the lower center of the image. These microcracks are attributed to thermal expansion mismatches or mechanical stresses induced during the sintering process. The surface of the sample appears rough and granular, which is typical of materials that have undergone high-temperature sintering. Moving to Figure 12B, the morphology is predominantly spherical, although occasional rods or polyhedral particles are also present in smaller numbers. The HAp samples exhibit distinct differences from the HApTi samples, such as rounded edges of the particles, lower porosity, and occasional triple junctions formed by the fusion of multiple particles. These features are likely a result of thermal treatment, as further illustrated in Figure 12C. In Figure 12D, the presence of nanometric HAp particles or agglomerates is evident. These particles are uniform in size, with diameters ranging between 70 and 120 nm.

Figure 13A–D show the SEM images taken on the surface of the HApTi samples. In Figure 13A the surface appears heterogeneous, having a granular texture where grains appear to be irregular in shape and size. Figure 13B reveals an irregular and porous structure, characteristic of composite materials where reinforcement particles are embedded within a ceramic matrix. At this higher magnification, the image shows a more detailed view of the distribution and morphology of Ti particles within the HAp matrix. The Ti particles appear as larger, brighter regions compared to the finer background structure of HAp (Figure 13C). In Figure 13D, it can be seen that, similar to the HAp samples, most of the particle diameters vary between 70 and 120 nm, but the presence of HAp nanometric particles or agglomerates with a diameter varying between 300 and 600 nm can be observed. The two-step sintering process employed in the fabrication of the implants plays a crucial role in maintaining the nanometric size of the HAp particles.

Using the SEM technique, we investigated the capability of the composites to act as substrates for the adhesion and proliferation of osteoblasts in vitro.

In Figure 14A, the texture is rough and granular, providing multiple potential adhesion points for cells. The prominent aggregate may indicate a region of cell adherence, but clear and distinct signs of cell presence, such as visible cell bodies or clusters, are not evident. Figure 14B reveals a more detailed view of the surface, showing fine particulate matter and irregularities, and several elongated structures can be observed, which might be remnants of cellular adhesion. In Figure 14C, larger aggregates of material are visible and cells appear to be well-adhered to the biomaterial, indicating good biocompatibility, and the presence of cellular protrusions (filopodia/lamellipodia) suggests active cellular interaction and movement. The images show varying stages of cell proliferation, from isolated clusters to dense networks, indicating healthy growth patterns, and the evidence of extracellular matrix production supports the notion of a supportive environment for cell growth. The overall morphology suggests a conducive environment for osteoblast adherence, with multiple attachment sites. In Figure 14D, visible clusters of MSCs are present, indicating that the cells are adhering and proliferating on the biomaterial surface. The texture appears granular, which may be due to the extracellular matrix or the interaction between the cells and the biomaterial. The MSCs display a spread-out morphology with some cells showing elongated shapes, which is typical for adherent cells indicating healthy growth. Cells seem to be well-adhered to the biomaterial, suggesting good biocompatibility of the substrate. There are surface protrusions or extensions (possibly filopodia) that indicate active cell movement and interaction with the environment.

The rough texture and particle distribution are supportive of cell growth, but direct signs of cell presence and proliferation are not discernible at this magnification (Figure 15A). In Figure 15B, the SEM image demonstrates successful cell adhesion, proliferation, and interaction with the biomaterial. The cells exhibit healthy growth characteristics, such as active protrusions, spreading, and extracellular matrix production. The Figure 15C SEM image provides clear evidence of osteoblast adherence and the initial stages of cell growth on the HApTi composite surface. The large adhered structures and visible filopodia extensions indicate active cell-surface interactions and robust attachment. The rough and irregular texture of the composite material supports the adhesion and spreading of osteoblasts, which is essential for bone tissue engineering applications. The observed cellular network formation suggests that the material has the potential to support further cell proliferation and tissue development. In Figure 15D, a dense layer of MSCs is observed, suggesting a high proliferation rate and confluency. There is evidence of extracellular matrix production, visible as a fibrous network, supporting cell attachment and growth. Cells show varied sizes and shapes, indicating different stages of cell growth and possibly differentiation. Similar to Figure 14D, there are protrusions indicating active cellular interactions, and clusters of cells are visible, which is a sign of proliferative activity. The experimental data indicate that both substrates can promote cellular proliferation and consequently tissue and bone regeneration.

### 3.2. EDS and XRD Results

SEM analysis of the sintered parts offered quantitative results, presented in Table 1 for three spectra (a, b, and c). The elemental composition analysis provided relevant information regarding the diffusion processes occurring during the sintering heat treatment. According to the thermophysical reaction occurring during TiH_2_ decomposition during sintering [19], the hydrogen evaporation determined voids formation and increased porosity, with TiH_2_, namely Ti, quantitatively decreasing (Table 1). On the other side, the oxygen content increased (Table 1) due to its reactivity with other chemical elements, and a lot of oxygen-based compounds were synthesized, as is confirmed by the XRD patterns in Figure 16.

Another advantageous aspect of the TSS heat treatment is the synthesis of TiO_2_ (Figure 17) as a rutile structure, which provides better osteoblast compatibility than other allotropic states of rutile (i.e., anatase) [20].

### 3.3. Biocompatibility Evaluation

The biocompatibility experiments conducted in this study aimed to analyze the effects of HAp and HApTi composites on MSC adhesion. The experiment involved cultivating cells for 5 days on the surface of the biocomposites, followed by cell fixation and fluorescent labeling to visualize components of the adhesion complex: actin in the cytoskeleton and vinculin in focal contact points. After scanning the entire surface of the samples using the automated TissueFAXSiPlus system, surface images were reconstructed to analyze cell density and distribution. It was observed that HAp pellet samples were populated by a small number of cells distributed randomly on the surface, whereas HApTi samples showed an increased number of adhered cells on the ceramic surface (Figure 18A). Cells grown on the test substrates formed a semi-confluent monolayer, with thin cell bodies and long, slender dendrites mediating intercellular communication. To visualize the cell–substrate interface in detail, cells were analyzed at a higher magnification (Figure 18B). The cytoskeleton of the cells grown on HAp and HApTi films was predominantly cortically distributed and connected to the substrate through bundled fiber groups forming predominantly peri-nuclear and few vinculin-mediated contacts (Figure 18C). The results indicate that HAp and HApTi composites have significant potential as substrates for use with osteoprogenitor cells. The cell culture-based biocompatibility analysis shows a minor phenotype improvement in HApTi samples over HAp. However, further in-depth research is needed to ascertain the biological importance of our observations.

### 3.4. In Vivo Study

#### 3.4.1. Computed Tomography Analysis

CT scanning revealed the presence of the osteoclastic process at the implant–bone interface and peripheral osteosclerosis, without any inflammatory process in the soft tissues. The CT analysis confirmed the correct placement of the implants during the study and an increase in bone density for the HApTi-implanted femur at two weeks and six weeks after implantation (Table 2). In the case of femurs implanted with HAp, all animals had a peri-implant physiological density of 800 ± 50 HU.

#### 3.4.2. Histological Analysis

In Figure 19A, the bone tissue is stained in varying shades of purple and pink, indicating the presence of different cellular and extracellular components. The bone structure appears trabecular with visible pores and spaces, indicative of cancellous bone. The large white area on the left side of the image represents the HAp implant. This non-biological material does not take up the hematoxylin–eosin stain, resulting in its appearance. In Figure 19B the interface between the bone and the HAp implant is clearly defined, showing areas of direct contact. The trabecular structure indicates ongoing bone remodeling and adaptation to mechanical stress. Osteocytes within the bone matrix are present, which are crucial for maintaining bone health and remodeling. Figure 19C displays the presence of adipose tissue near the bone–implant interface because the implant is located near or within the marrow cavity, which contains fat cells. The interface does not show active new bone formation or remodeling, no fibrosis, and no inflammation. The adipose tissue appears healthy, with no signs of inflammation or necrosis, indicating a stable environment around the implant. The implant material remains unstained and distinct from the surrounding tissue. Dark spots within the white area could represent artifacts or remnants of the implant material. Chronic inflammation elements (macrophages, foreign body giant cells, or neutrophils) and necrosis elements could not be detected eight weeks post-implantation (Figure 19A–C).

While Figure 20A does not show direct bone–implant interaction, the presence of healthy connective tissue and the lack of inflammation suggest a stable environment. This stage may represent an early phase of healing, where connective tissue precedes bone formation. In Figure 20B the bone tissue appears healthy, with no signs of inflammation or necrosis. The implant material remains distinct from the surrounding tissue, suggesting no adverse reaction. The interface between the bone and the implant material is well-defined, showing areas of close contact. In Figure 20C no signs of inflammation or necrosis are visible, and the bone–implant interface shows direct contact and no significant gaps. Chronic inflammation indicators, including macrophages, foreign body giant cells, neutrophils, and necrosis, were absent eight weeks post-implantation (Figure 20A–C). The absence of osteointegration elements visible in the histological analysis may be due to the short period the implant was maintained in the femurs of the animals in this study.

#### 3.4.3. Immunohistochemical Analysis

Examination of immunohistochemical sections for OC was performed both in transmitted light to visualize osteoblasts and non-collagen proteins (A, C, E, G) and in polarized light to visualize birefringent collagen fibers (B, D, F, H).

To observe the osteointegration of the two types of implants and to highlight their osteoformative potential, we conducted an immunohistochemical analysis. In the femurs implanted with HAp, osteocalcin (OC) was expressed in the extracellular bone matrix (Figure 21 and Figure 22). Transmitted light microscopic analysis revealed the presence of implant fragments embedded in the extracellular matrix and detached from the bone, osteoblasts, and areas of the implant that adhered to the bone. From the polarized light images, implant adhesions to bone, areas with OC without birefringence, and the absence of birefringent collagen bundles in the implant incorporation zone were observed. In the femurs implanted with HApTi, OC was expressed in osteoblasts, Haversian canals, and the bone matrix. Utilizing transmitted light microscopy, we detected the attachment of the implant to the bone tissue surface, integration of implant fragments into the bone mass, and the formation of new collagen fibers. Images obtained using polarized light indicated the existence of birefringent collagen fibers, which facilitate the incorporation of implant fragments within the bone integration zone, as well as the emergence of newly developed collagen fibers (Figure 21 and Figure 22). The examination performed at the bone–implant interface for both categories of tested implants shows zones of implant attachment, implant particles integrated into the bone matrix, birefringent collagen fibers, and the presence of osteoblasts. Despite both types of implants exhibiting a satisfactory level of osteointegration, several factors indicated the superior osteointegration of the HApTi implants.

#### 3.4.4. Histomorphometric Analysis and Statistical Analysis

The determination of bone density through histomorphometric analysis showed a higher value for HApTi (0.00008128/μm^2^) compared to femurs implanted with HAp (0.00007236/μm^2^) (Table 3). The coefficient of variation was higher in femurs implanted with HApTi (15.2664%) compared to those implanted with HAp (10.5987%). The results of the paired-sample *t*-test for comparing the average number of osteoblasts/μm^2^ did not show statistically significant differences between the left and right femurs (*p* > 0.05 (0.086928948); degrees of freedom < 200 (3)) (Table 4).

## 4. Discussion

The two-step thermal sintering treatment is a promising approach for simultaneously obtaining composites with a nanometric structure of the component particles. The method of two-step thermal sintering treatment involves constraining particle growth while simultaneously allowing particle diffusion [21,22,23].

In a study by Cândido et al., the histological analysis of the interfaces between bone structure and heteroplastic implants in the tibias of dogs was presented. The implants were maintained in the animal’s tibias for eight months, after which the animals were sacrificed and the samples processed for analysis. Microscopic analysis highlighted a large amount of osteoid tissue, osteoblasts, and osteocytes at the bone–implant interface, without the presence of fibrous tissue [24]. In a study conducted by Bumbu et al., it was shown that osteointegration of Ti implants was complete in half of the animals included in this study two months post-implantation. Osteointegration and immobilization of the implant were achieved by forming a very thin layer of bony callus at the bone–metal implant junction [25].

The in vivo behavior of HapTi-based biocomposites is also explained by the effect of the TSS-type sintering treatment on the properties. During the sintering process, TiH_2_ decomposes and releases hydrogen gas. This occurs in stages where TiH_2_ converts into TiHx and α-Ti. This released hydrogen creates additional pore spaces within the composite material, increasing the porosity of the HApTi samples. As demonstrated in a study by Marinescu C. et al., by means of the TSS technological parameters used, homogeneous biocomposite structures of HApTi were obtained, with a pore size of 200–600 nm and an obvious improvement in the physical-mechanical properties, which determined superior osseointegration compared to tablets from HAp (with a porosity value for pores smaller than 125 nm) [26]. Hu F. et al. investigated the synthesis and characterization of porous hydroxyapatite using titanium hydride. Their study demonstrated that the porosity significantly enhanced the mechanical properties and bioactivity compared to non-porous hydroxyapatite [27]. A research team focused on the fabrication of porous hydroxyapatite using titanium oxide as a pore-forming agent. Their results showed improved osteointegration and cell proliferation in the porous samples compared to their non-porous counterparts [28]. A study by Amirnejad M. et al. explored the effects of incorporating titanium dioxide into hydroxyapatite to create a porous structure. Their findings highlighted the superior compressive strength and biological performance of the porous hydroxyapatite over the non-porous form [29]. The work of Boyne P. et al. involved a comparative study on biological responses to porous and non-porous hydroxyapatite. The porous hydroxyapatite, fabricated with titanium hydride, exhibited enhanced vascularization and tissue ingrowth [30]. Other researchers synthesized porous hydroxyapatite with titanium oxide and compared its mechanical and biological properties with non-porous hydroxyapatite. They found that the porous structure facilitated better nutrient transport and cellular attachment [31]. Research conducted by Lee H. et al. focused on the drug delivery capabilities of hydroxyapatite. By using titanium oxide to create porosity, they demonstrated that the porous hydroxyapatite had a higher drug loading capacity and controlled release profile compared to non-porous samples [32]. In their investigation, Niespodziana K. et al. studied the thermal properties of porous hydroxyapatite made with titanium hydride. Their comparative analysis with non-porous hydroxyapatite revealed that the porous material had improved thermal stability and resistance to thermal degradation [33]. Stich T. et al. evaluated the osteoconductivity of porous hydroxyapatite synthesized using titanium hydride. The porous samples showed significantly higher bone formation and mineralization rates compared to non-porous hydroxyapatite [34]. Ielo I. et al. conducted an in vivo performance study of porous hydroxyapatite with titanium oxide. Their study demonstrated that the porous hydroxyapatite promoted faster and more complete healing of bone defects than non-porous hydroxyapatite [35]. Koju N. et al. investigated the degradation rates of porous and non-porous hydroxyapatite. By using titanium oxide to induce porosity, they found that the porous hydroxyapatite degraded at a more controlled and predictable rate, which is beneficial for bone regeneration applications [36].

## 5. Conclusions

The two-step thermal sintering process proved effective in producing biocomposites with controlled microstructures and enhanced mechanical properties. SEM revealed that HApTi samples exhibited a more heterogeneous and porous structure, with improved particle distribution compared to HAp samples. In vitro studies demonstrated that both HAp and HApTi biocomposites supported the adhesion and proliferation of MSCs; however, HApTi showed a slight advantage, as evidenced by fluorescence microscopy and image cytometric analysis. CT scans indicated a higher peri-implant bone density for HApTi implants at both two and six weeks post-implantation. Histological analysis confirmed these findings, showing healthy bone tissue with no signs of fibrosis or inflammation around both types of implants. Notably, HApTi implants exhibited a denser and more organized bone structure at the interface, indicating superior osteointegration. Immunohistochemical analysis further supported these observations, with HApTi implants showing higher expression of OC in osteoblasts and the surrounding bone matrix. Histomorphometric analysis quantified these qualitative observations, showing a higher number of osteoblasts per square micrometer around HApTi implants compared to HAp implants. While both types of implants demonstrated good osteointegration, the statistical analysis underscored the superior performance of HApTi in promoting bone cell activity and integration. The study conclusively demonstrates that HApTi biocomposites offer significant advantages over pure HAp in terms of structural properties, biocompatibility, and osteointegration. These findings suggest that HApTi implants are more suitable for clinical applications in orthopedic surgery, particularly in scenarios requiring rapid and robust bone healing and stability.

## Figures and Tables

**Figure 1 jfb-15-00181-f001:**
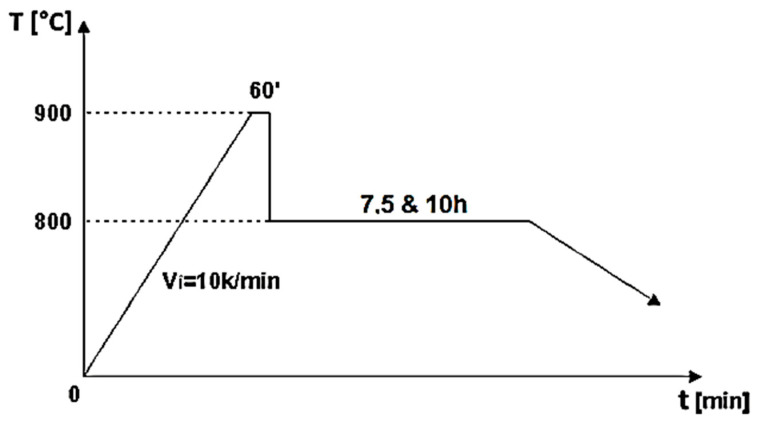
Two-step sintering cycle for the biocomposite samples [8].

**Figure 2 jfb-15-00181-f002:**
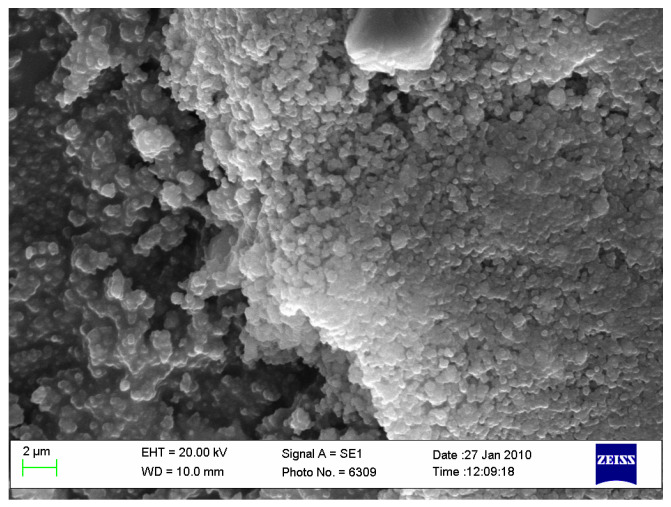
Biocomposite powder mixture of HAp with titanium hydride (ourtesy of Dr. Gingu Oana, University of Craiova, Romania).

**Figure 3 jfb-15-00181-f003:**
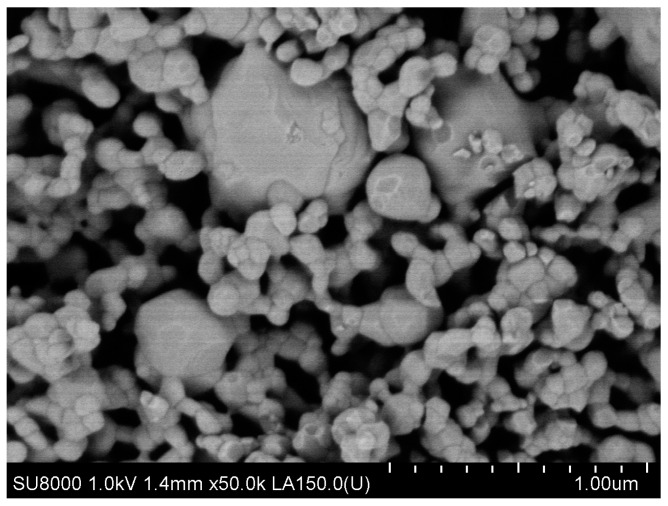
Sintered biocomposite state [9].

**Figure 4 jfb-15-00181-f004:**
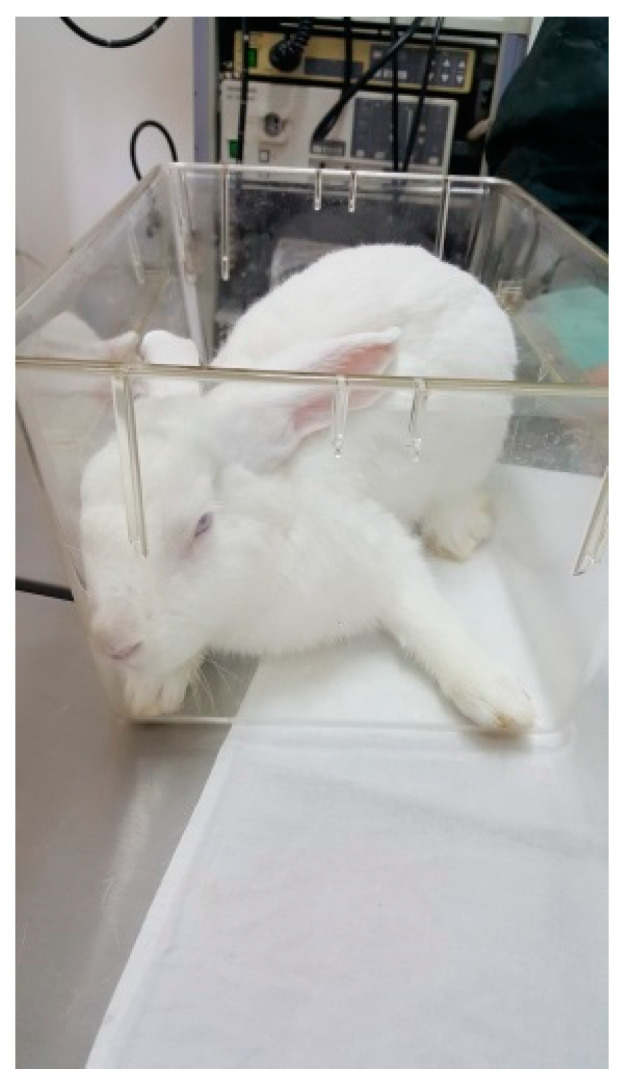
A New Zealand White rabbit.

**Figure 5 jfb-15-00181-f005:**
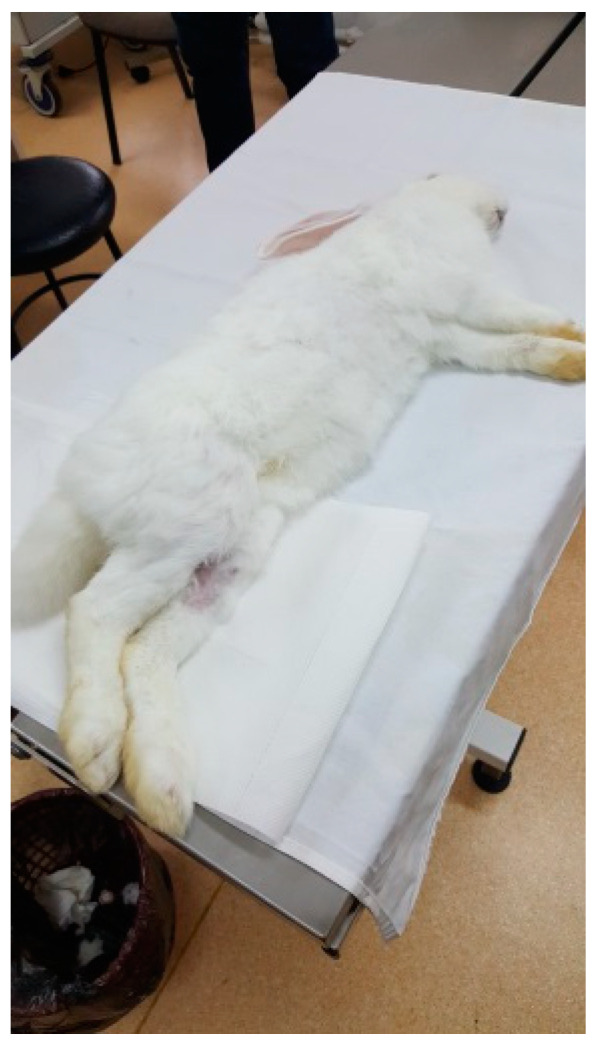
Anesthetized rabbit in the operating area.

**Figure 6 jfb-15-00181-f006:**
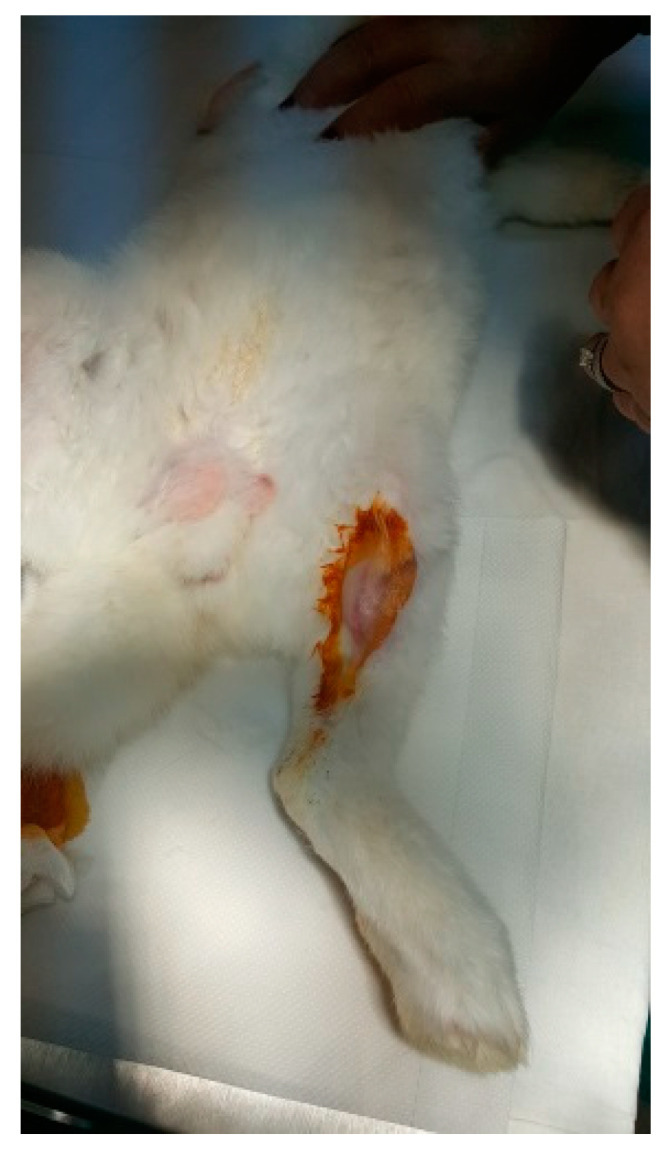
Shaved and disinfected incision area.

**Figure 7 jfb-15-00181-f007:**
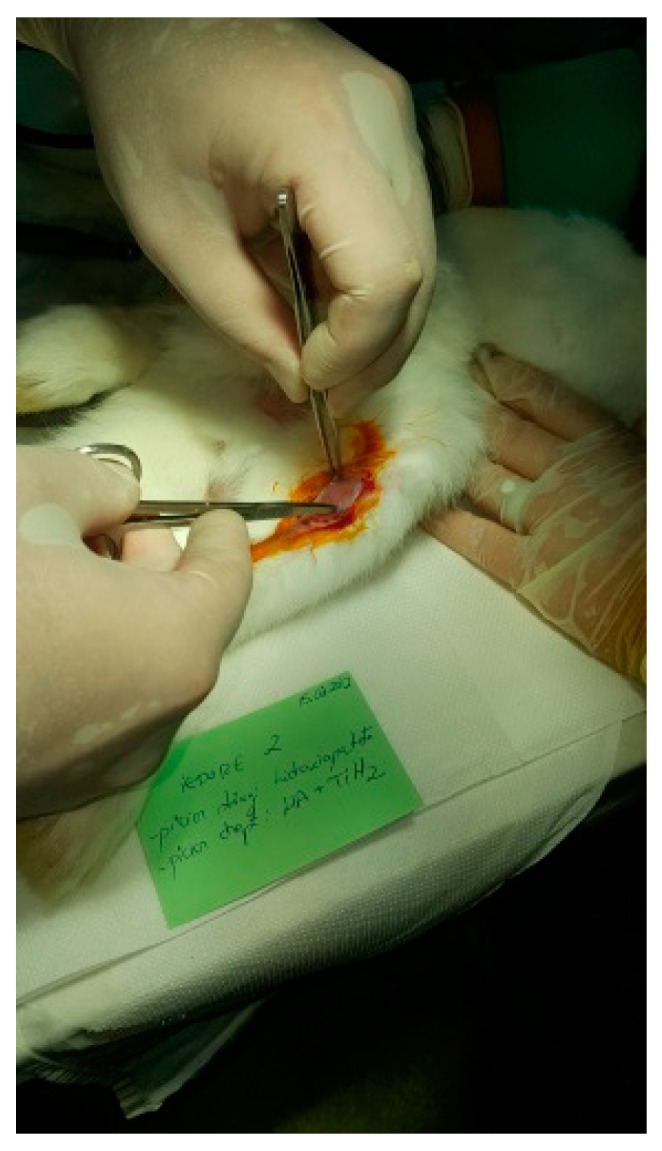
Tissue incision.

**Figure 8 jfb-15-00181-f008:**
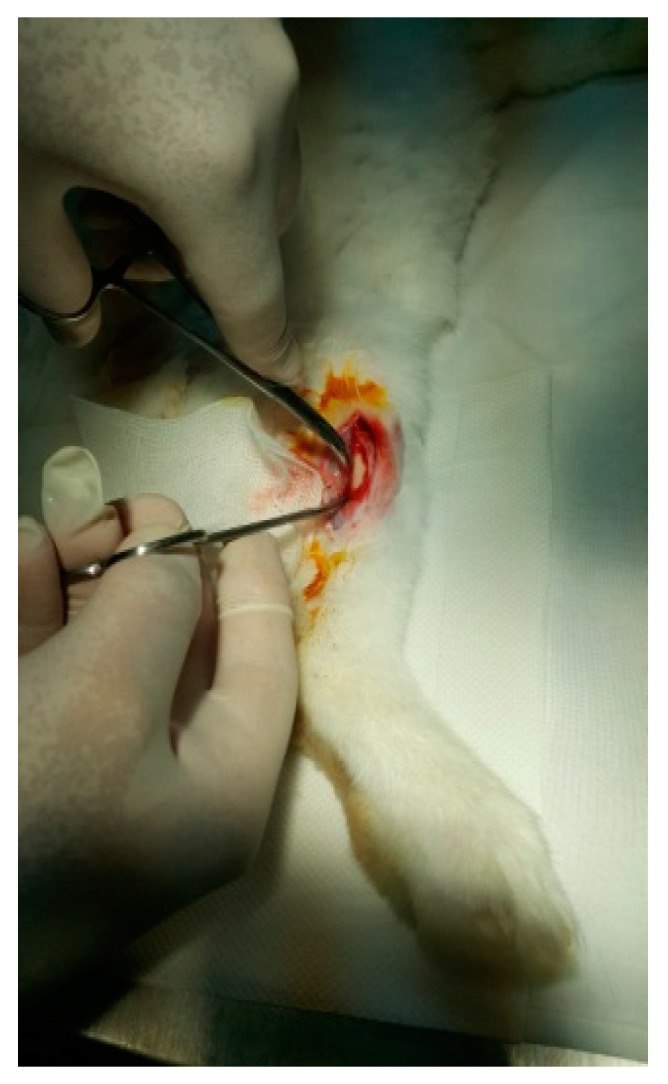
Exposure of the femur.

**Figure 9 jfb-15-00181-f009:**
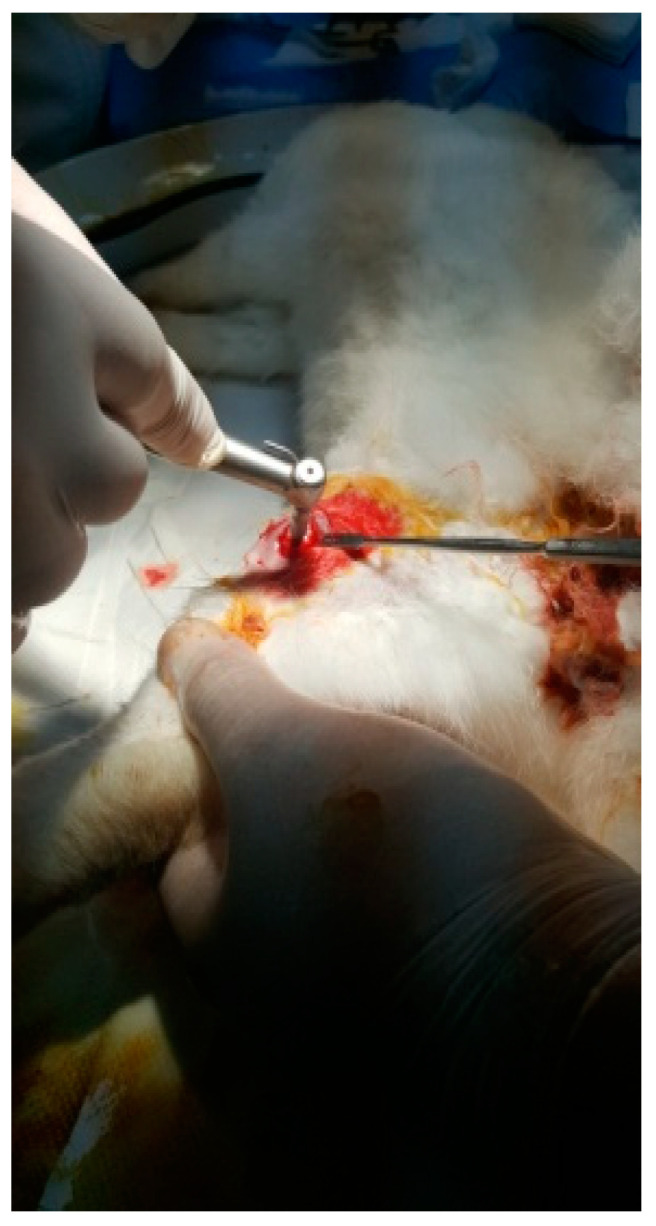
Incision of the femur.

**Figure 10 jfb-15-00181-f010:**
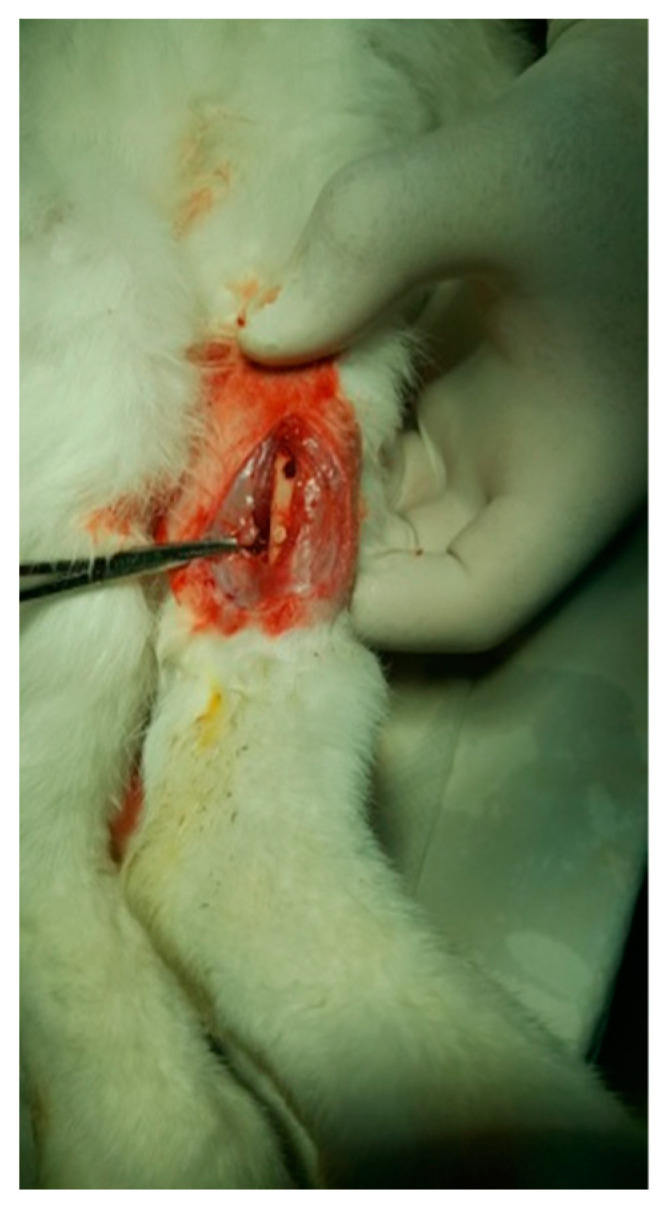
Fixation of the implant.

**Figure 11 jfb-15-00181-f011:**
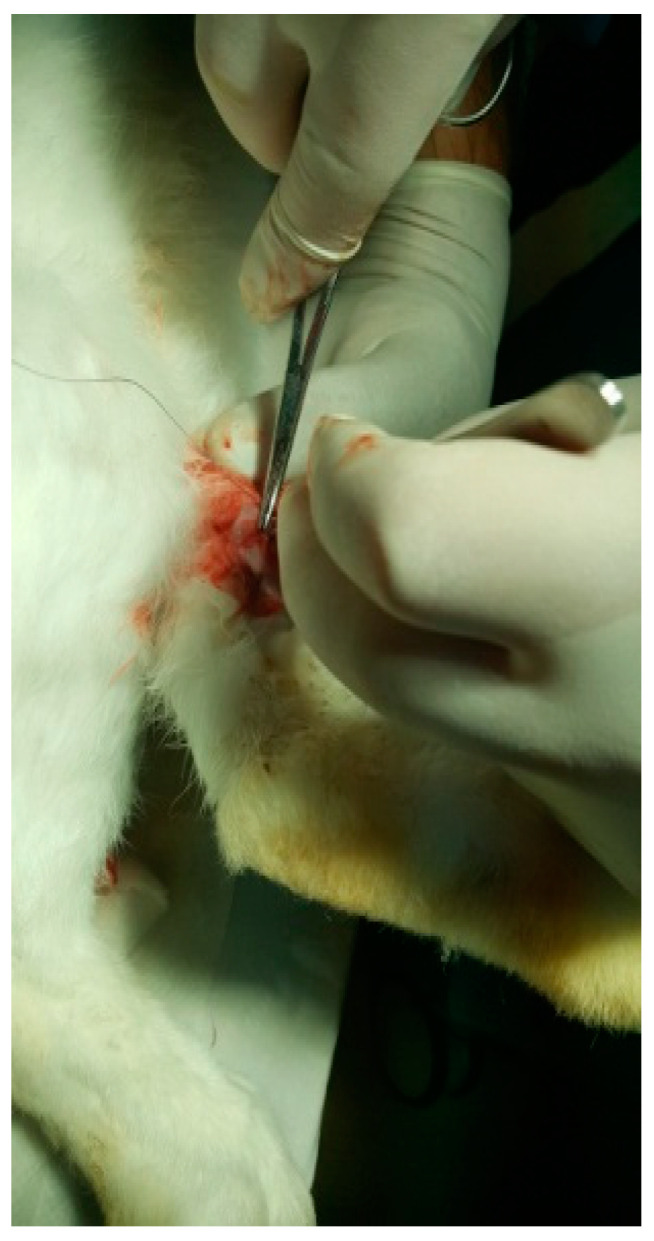
Suturing the incision.

**Figure 12 jfb-15-00181-f012:**
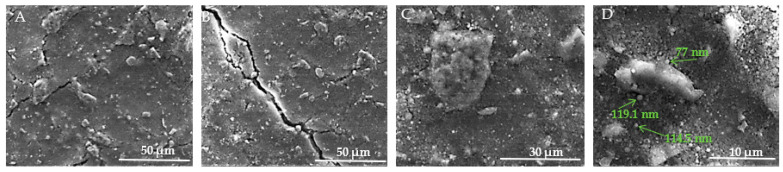
SEM images of HAp implants. (**A**) Visible microcracks extending across the surface, attributed to thermal expansion mismatches or mechanical stresses during sintering, with a rough and granular texture at 50 μm scale. (**B**) Predominantly sphericalorphology with occasional rods or polyhedral particles, showing rounded edges, lower porosity, and triple junctions formed by particle fusion at 50 μm scale. (**C**) Detailed view of particle morphology, further illustrating thermal treatment effects at 30 μm scale. (**D**) Presence of nanometric HAp particles or agglomerates, uniform in size with diameters ranging between 70 and 120 nm, at 10 μm scale. Arrows indicate measurements of notable particles.

**Figure 13 jfb-15-00181-f013:**
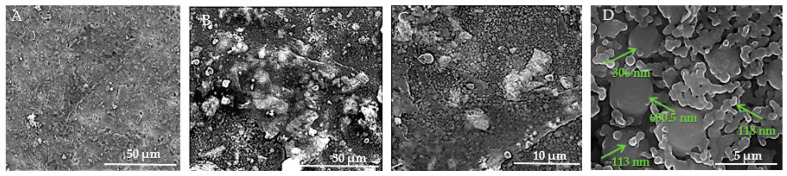
SEM images of HApTi implants. (**A**) Heterogeneous surface with granular texture and irregularly shaped and sized grains at 50 μm scale. (**B**) Irregular and porous structure characteristic of composite materials with Ti particles embedded within the HAp matrix at 30 μm scale. (**C**) Distribution and morphology of Ti particles within the HAp matrix, with Ti particles appearing as larger, brighter regions against the finer HAp background at 10 μm scale. (**D**) Nanometric particles and agglomerates, with particle diameters varying between 70 and 120 nm, and some HAp particles or agglomerates between 300 and 600 nm, at 5 μm scale. Arrows indicate measurements of notable particles.

**Figure 14 jfb-15-00181-f014:**
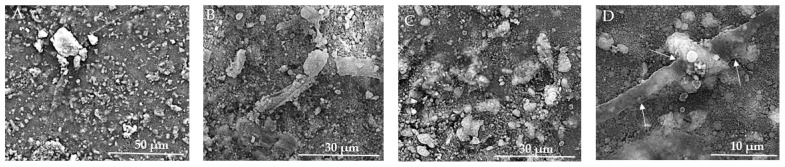
SEM images of HAp implants after the in vitro study. (**A**) Rough, granular texture with potential adhesion points for cells at 50 μm scale. (**B**) Fine particulate matter and irregularities with elongated structures at 30 μm scale. (**C**) Larger aggregates and well-adhered cells with cellular protrusions (filopodia/lamellipodia) at 30 μm scale. (**D**) Clusters of MSCs with granular texture and surface protrusions/extensions indicating cell interaction at 10 μm scale. Arrows indicate surface protrusions or extensions (possibly filopodia).

**Figure 15 jfb-15-00181-f015:**
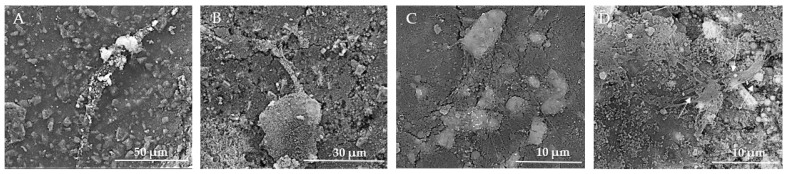
SEM images of HApTi implants after the in vitro study. (**A**) Rough texture and particle distribution supportive of cell growth but without discernible direct signs of cell presence at 50 μm scale. (**B**) Successful cell adhesion and proliferation with active protrusions and extracellular matrix production at 30 μm scale. (**C**) Clear evidence of osteoblast adherence and initial cell growth stages, with large adhered structures and visible filopodia extensions indicating robust attachment at 10 μm scale. (**D**) Dense layer of MSCs indicating high proliferation rate and confluency, with extracellular matrix production and varied cell shapes at 10 μm scale. Arrows indicate notable features such as cellular protrusions and clusters.

**Figure 16 jfb-15-00181-f016:**
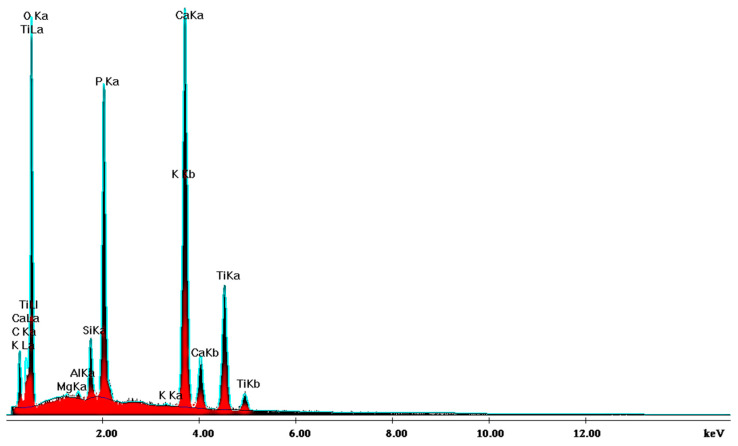
XRD pattern providing the elemental composition of the samples in sintered state.

**Figure 17 jfb-15-00181-f017:**
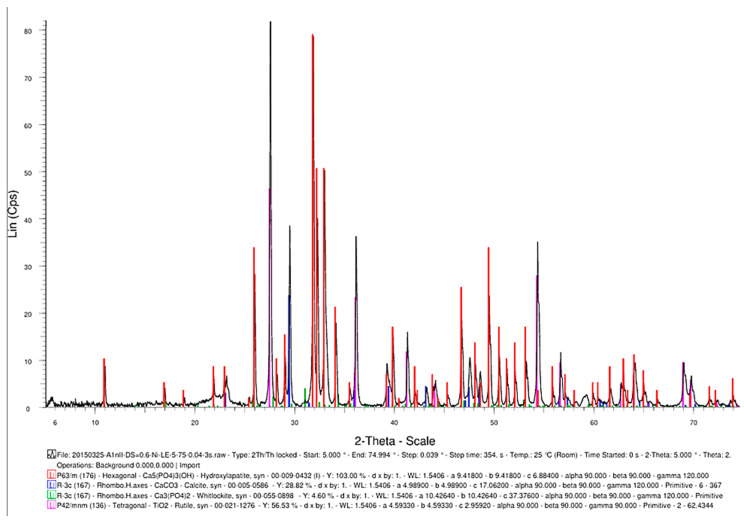
The phase composition of the sintered samples, showing the synthesis of hydroxyapatite, CaCO_3_, Ca_3_(PO_4_)_2_, and TiO_2_.

**Figure 18 jfb-15-00181-f018:**
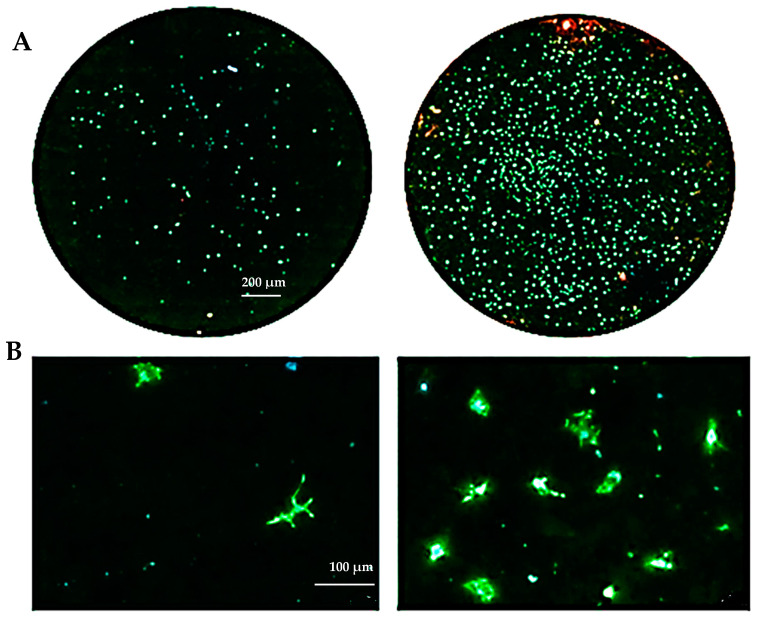
Adhesion of MSCs 5 days post-cultivation on test materials HAp (left) and HApTi (right). The entire specimen was reconstructed by scanning the samples using the TissueFAXSiPlus system (**A**). A representative individual field obtained with the microscope’s 20× objective is shown (**B**). Detailed images of the cells can be observed at 40× magnification using the Zeiss Axio Imager microscope. Actin was highlighted using Alexa Fluor 488 Phalloidin (green), vinculin with Alexa Fluor 594 (red), and nuclei were stained with Hoechst (blue) (**C**).

**Figure 19 jfb-15-00181-f019:**
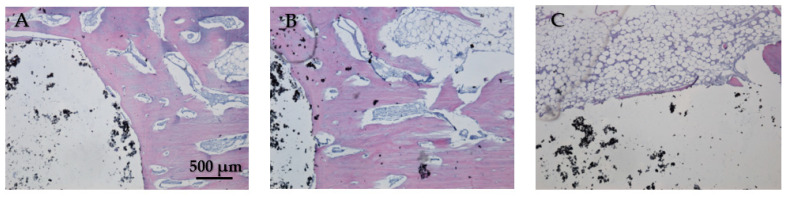
Cross-sectional view of the femur implanted with HAp after two months of implantation. Staining of the smear was performed with hematoxylin–eosin mixture, 4× magnification. (**A**) Trabecular bone structure with HAp implant visible as a large white area. (**B**) Clear bone-implant interface with direct contact and ongoing bone remodeling. (**C**) Presence of healthy adipose tissue near the bone-implant interface with no signs of inflammation or necrosis.

**Figure 20 jfb-15-00181-f020:**
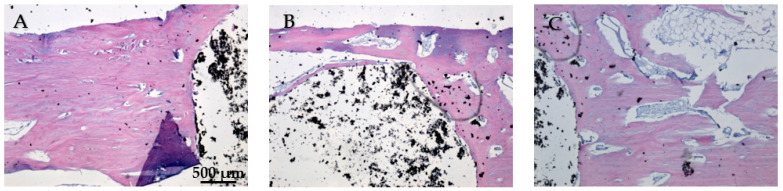
Cross-sectional view of the femur implanted with HApTi after two months of implantation. Staining of the smear was performed with hematoxylin–eosin mixture, 4× magnification. (**A**) Healthy connective tissue with no inflammation, indicating a stable environment. (**B**) Healthy bone tissue with a well-defined bone-implant interface and no adverse reactions. (**C**) Direct bone-implant contact with no inflammation or necrosis, indicating good biocompatibility.

**Figure 21 jfb-15-00181-f021:**
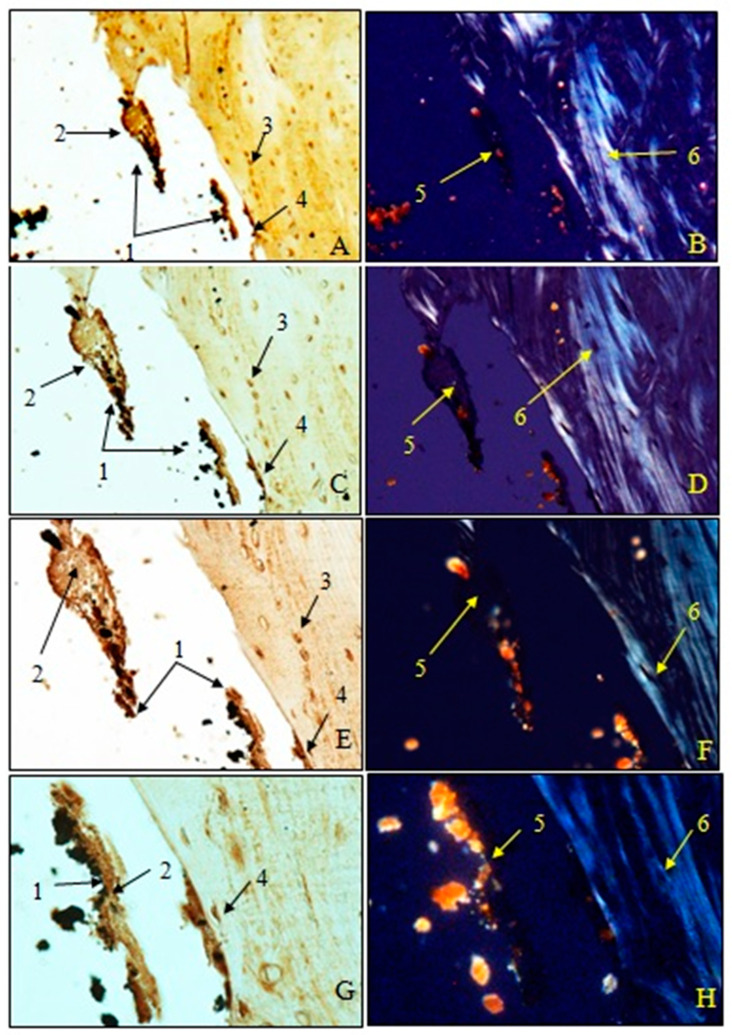
Morphogenesis of implant integration into the adjacent bone tissue. Femur implanted with HAp (Osteocalcin). 1. Fragments of implant incorporated into the extracellular matrix and detached from the bone. 2. Extracellular matrix of the bone containing OC. 3. Osteoblasts. 4. Implant adhered to bone. 5. Zone with OC without birefringence upon examination under polarized light. 6. Bundles of birefringent collagen fibers upon examination under polarized light. Magnification: ×28 (**A**,**B**); ×140 (**C**,**D**); ×210 (**E**,**F**); ×280 (**G**,**H**).

**Figure 22 jfb-15-00181-f022:**
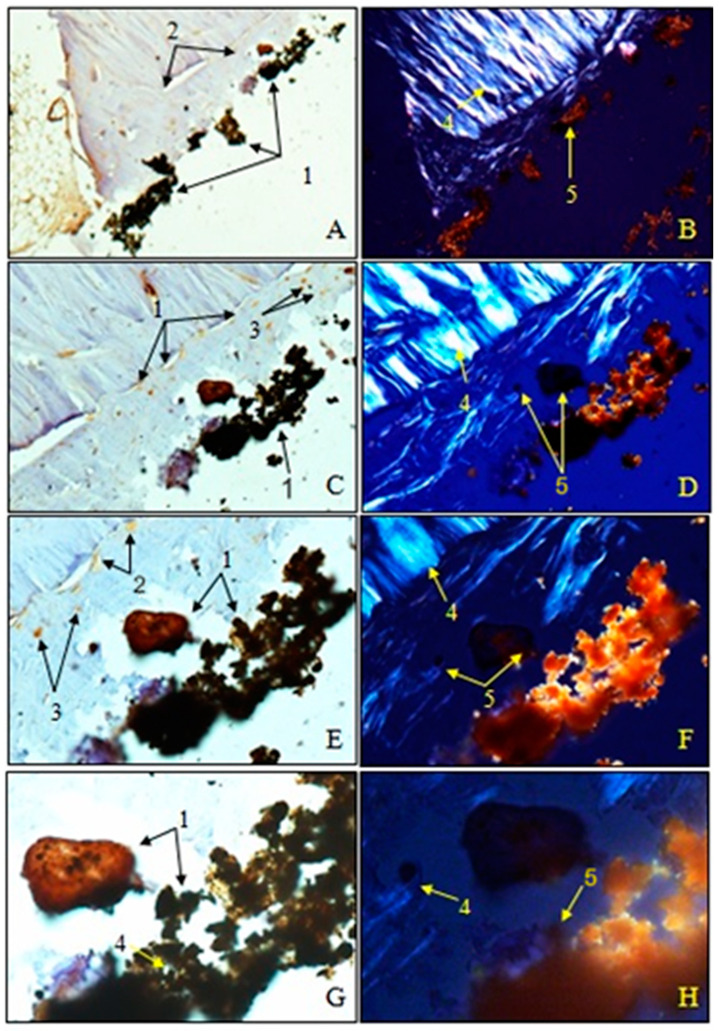
Morphogenesis of implant integration into the adjacent bone tissue. Femur implanted with HApTi (Osteocalcin). 1. Fragments of the implant incorporated into the bone extracellular matrix. 2. OC in the bone extracellular matrix. 3. Osteoblasts. 4. Bundles of birefringent collagen fibers upon examination under polarized light. 5. Absence of birefringent collagen fiber bundles in the implant incorporation zone. Magnification: ×28 (**A**,**B**); ×140 (**C**,**D**); ×210 (**E**,**F**); ×280 (**G**,**H**).

**Table 1 jfb-15-00181-t001:** EDS analysis of the sintered samples.

	a	b	c	AVERAGE	SD	%RSD
**CK**	11.49	12.39	14.3	12.73	1.43	11.28
**OK**	57.6	59.56	58.27	58.48	1.00	1.7
**MgK**	0.08	0.11	0.07	0.09	0.02	24.02
**PK**	8.28	8.14	7.86	8.09	0.21	2.64
**CaK**	15.68	14.12	13.67	14.49	1.05	7.28
**TiK**	6.86	5.68	5.83	6.12	0.64	10.49
	100	100	100	100		
**Ca/P**	1.8937	1.7246	1.7391	1.79	0.09	5.06
**Ti/Ca**	0.4375	0.4022	0.4264	0.42	0.02	4.27
**Ti/P**	0.8285	0.6977	0.7417	0.76	0.07	8.8

**Table 2 jfb-15-00181-t002:** Bone density measurement at the implant–bone interface, expressed using U.H. (Hounsfield Units).

Femurs	At Two Weeks	At Six Weeks	Description of Implant–Bone Interface
Left femurs	792–903 U.H.	574–978 U.H.	Osteoclastic activity present, peripheral osteosclerosis observed, no inflammatory process in the soft tissues.
Right femurs	625–1051 U.H.	1082–1512 U.H.	

**Table 3 jfb-15-00181-t003:** Results of the paired *t*-test for comparing the number of osteoblasts/μm^2^ for left femur implanted with HAp and right femur implanted with HApTi.

Femur Implanted With	Mean	Standard Deviation	Coefficient of Variation
HAp	0.00007236	0.0000102	10.5987%
HApTi	0.00008128	0.0000124	15.2664%

**Table 4 jfb-15-00181-t004:** Paired *t*-test results for comparing the number of osteoblasts/μm^2^ in Group I (left femurs implanted with HAp and right femurs implanted with HApTi).

*t*-Statistic	Degrees of Freedom	*p*
1.884369571	3	0.086928948

## Data Availability

The original contributions presented in the study are included in the article, further inquiries can be directed to the corresponding authors.

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
