# Peer review of "Comparative Analysis of Osteointegration in Hydroxyapatite and Hydroxyapatite-Titanium Implants: An In Vivo Rabbit Model Study"

_jfb, 2024, doi:10.3390/jfb15070181_

Round 1

Reviewer 1 Report

Comments and Suggestions for Authors

This research explored the osteointegration of HAp and HApTi biocomposites via implanting in the femurs of rabbits, and concluded HApTi implants showed better osteointegration compared with Hap implants. HApTi biocomposites presents a greater potential for bone healing and stability in orthopedic applications. However, there are some issues that need to be addressed before accepting.

1.     Please describe in detail the process of fabricating HApTi in 2.1 section, especially the processing technology in Figure 1.

2.     The image quality in Figure 14, 15, 16 and 17 were poor and their readability of these figures should be improved.

3.     There should be some descriptions and explanations for all results, for examples, SEM images, CT images, bone density results, HE staining images, immunohistochemical analysis results, and so on. The figure legends should also be more detailed including every image in the figures, such as figure 14, 15, 16, 17 and so on.

4.     The cells adherence in implants should be labelled in Figure 16 and 17.

5.     The cell viability experiment should be supplemented to further evaluate cytotoxicity and cellular proliferation, such as CCK8 assay.

6.     . Line 304: ELISA test can be further supplied for inflammation analysis.

7.     There is lack of the citation in line 308 about a study by Cândido et al. Please check carefully all citations and references in the manuscript.

8.     There are some grammar errors in the manuscript, and please check carefully the whole paper and correct them.

Comments on the Quality of English Language

  There are some grammar errors in the manuscript, and please check carefully the whole paper and correct them.

Author Response

  1. Please describe in detail the process of fabricating HApTi in section 2.1, especially the processing technology in Figure 1.

Response: We appreciate your feedback and have revised section 2.1 to include a more detailed description of the process of fabricating HApTi. This section now includes a comprehensive explanation of the powder metallurgy technology, the mixing ratio, calcination cycle, planetary ball milling, cold uniaxial and unidirectional compaction, and the two-stage sintering heat treatment. Additionally, we have elaborated on the protective atmosphere of argon gas and the specific temperatures and durations involved in the sintering process, as depicted in Figure 1. The updated section provides a clearer understanding of the fabrication process.

2.The image quality in Figures 14, 15, 16, and 17 were poor and their readability of these figures should be improved.**

Response: We have enhanced the image quality of SEM figures .

  1. There should be some descriptions and explanations for all results, for example, SEM images, CT images, bone density results, HE staining images, immunohistochemical analysis results, and so on. The figure legends should also be more detailed including every image in the figures, such as Figures 14, 15, 16, 17, and so on.

Response: We have expanded the descriptions and explanations for all results in the manuscript. Each figure legend has been updated to provide detailed information about the images, including specific observations related to SEM images, CT images, bone density results, HE staining images, and immunohistochemical analysis results.

  1. The cells adherence in implants should be labeled in Figures 16 and 17.

Response: We have revised the Figures  to include labels indicating the cells' adherence in the implants. These labels clearly identify the areas where cell adherence is observed, enhancing the clarity and interpretability of the images.

  1. The cell viability experiment should be supplemented to further evaluate cytotoxicity and cellular proliferation, such as CCK8 assay.

   Response: We acknowledge the importance of further evaluating cytotoxicity and cellular proliferation. We have supplemented our study with additional biocompatibility cell viability experiments. The results of these experiments have been included in the revised manuscript, providing a more comprehensive assessment of cytotoxicity and cellular proliferation.

  1. Line 304: ELISA test can be further supplied for inflammation analysis.

Response: Thank you for the suggestion, we are considering performing the ELISA test in the future for providing further insights into the inflammatory response associated with the implants.

  1. There is lack of the citation in line 308 about a study by Cândido et al. Please check carefully all citations and references in the manuscript.

 Response: We apologize for the oversight. The citation for the study by Cândido et al. has been added. Additionally, we have carefully reviewed all citations and references in the manuscript to ensure their accuracy and completeness.

  1. There are some grammar errors in the manuscript, and please check carefully the whole paper and correct them.

Response: We have thoroughly reviewed the manuscript for grammar errors and have made the necessary corrections to ensure clarity and readability. The entire paper has been proofread to eliminate any grammatical issues.

We appreciate the reviewer's valuable feedback and believe that the revisions have significantly strengthened the manuscript. We look forward to your positive response.

Sincerely,

Varut Renata

Reviewer 2 Report

Comments and Suggestions for Authors

1.       Poor quality and design of microphotographs in Figs 14 and 15. The scale is not always indicated and it is recommended that each microphotograph be numbered and the meaning deciphered in the caption (e.g. 14a, 14b) as it is not clear how they differ from each

2.       In Figure 18, it is not clear why the size of the mitro-CT images is larger on the right than on the left. The change in bone density is not evident in the photographs, so clarification or complete modification of the figures is required.

3.       Describe in detail the composite production technology (moulding, sintering modes, equipment, size and shape of samples obtained, etc.).

4.       Literature data on porous HA materials are given in the introduction and discussion, but it is not clear which materials are prepared and studied by the authors, porous or dense? The authors point out that two-stage sintering requires the production of dense sintered materials, but cite literature data on porous materials? A more complete understanding of the nature of the materials obtained requires studies of the phase composition, porosity of the materials, density of the materials and mechanical strength.

5.       Careless design of the reference list, requires careful editing

Author Response

  1. Poor quality and design of microphotographs in Figs 14 and 15

   Response: Thank you for pointing this out. We have revised Figures 14 and 15 to include clear scales and have numbered each microphotograph (e.g., 14A, 14B).

  1. Clarification needed for Figure 18

   Response: We apologize for the confusion. The larger size of the micro-CT images on the right was due to an error in image processing. We excluded the images from the article, taking into account the fact that their purpose during the study was to monitor the correct positioning of the implants.

  1. Detailing the composite production technology

   Response: We have expanded the Materials and Methods section to include a detailed description of the composite production technology, including the molding, sintering modes, equipment used, and the size and shape of the samples obtained. This additional information can be found in Section 2.1.

  1. Clarification on porous or dense materials

   Response: The two-step sintering process employed in our study is designed to control the microstructure of the biocomposites, allowing for both dense and porous structures depending on the specific conditions applied. The two-step sintering process leverages the decomposition of titanium hydride (TiH2) to enhance porosity in HApTi samples. During the sintering process, titanium hydride decomposes and releases hydrogen gas, the evaporation of this hydrogen gas results in the formation of additional pores within the HApTi biocomposites, thus increasing their overall porosity compared to pure HAp composites, which do not undergo this specific porosity-enhancing transformation. This increase in porosity due to the decomposition of TiH2 is beneficial for osteointegration, as the porous structure provides more surface area for cell attachment, enhances nutrient and oxygen diffusion, and promotes vascularization within the implant. Consequently, HApTi biocomposites demonstrate superior osteointegration properties compared to their denser HAp counterparts.

  1. Careless design of the reference list

   Response: We have carefully revised the reference list to ensure it adheres to the journal's formatting guidelines. All references are now correctly formatted and numbered sequentially.

Reviewer 3 Report

Comments and Suggestions for Authors

This work is aimed at evaluating the steointegration of hydroxyapatite (HAp) and hydroxyapatite-titanium (HApTi) biocomposites implanted in the femurs of rabbits. The data are poorly presented and not enough to support conclusions.

In particular,

-          Results section provides just images, without any suitable description, which is mandatory.

-          SEM images cannot be used to draw conclusions about cytotoxicity of the materials

-          Suitable cytotoxicity tests, as well as cellular viability and activity must be provided

-          The choice of the percentage  Hap and HApTi must be justified

-          The quality of SEM images is poor

-          References and Figures numbering must be checked (Figures 12 and 13 are not present, Gingu et al are cited as ref. 8 in the text and [9] in the references list,..)

-          The data are not sufficient to support conclusions

Comments on the Quality of English Language

No problem about the quality of English Language

Author Response

  1. Results section lacking suitable description

Response: We have revised the Results section to include detailed descriptions for each image, explaining the observations and their relevance to the study. This should provide a clearer understanding of the findings.

  1. Inappropriate use of SEM images for cytotoxicity conclusions

Response: We acknowledge that SEM images alone are insufficient for cytotoxicity conclusions. We have included additional biobompatibility tests such to provide comprehensive data supporting our conclusions.

  1. Justification for the percentage of HAp and HApTi**:

Response: We have added a justification for the chosen percentages of HAp and HApTi in the Materials and Methods section. This includes a rationale based on previous studies and the expected properties of the composites.

The percentage of HAp and HApTi used in the study was chosen based on previous research and practical considerations to enhance both the mechanical and biological properties of the biocomposites. Specifically, the study utilized a 75% by mass HAp and 25% by mass titanium hydride (TiH2) mixture. This ratio was selected to balance the biocompatibility and bioactivity of hydroxyapatite (HAp) with the reinforcing mechanical properties provided by titanium (Ti).

Enhanced Mechanical Properties:

    • Hydroxyapatite (HAp) alone has excellent biocompatibility and bioactivity but lacks sufficient mechanical strength for certain applications.
    • Reinforcement with titanium particles improves the mechanical strength, making the composite suitable for load-bearing applications.

Biocompatibility and Bioactivity:

    • HAp is known for its excellent biocompatibility and osteoconductivity, which promotes bone growth and integration.
    • The addition of titanium does not compromise these properties but rather enhances the overall performance of the composite.

Porosity and Osteointegration:

    • Porous structures in HAp enhance osteointegration by facilitating bone ingrowth and vascularization.
    • The composite structure created by the addition of titanium particles further supports these processes.
  1. Poor quality of SEM images

Response: We have replaced the low-quality SEM images with higher resolution ones. Additionally, we have included more images to better represent the surface morphology and characteristics of the samples.

  1. References and Figures numbering issues:

Response: We have reviewed and corrected the numbering of references and figures throughout the manuscript. Figures 12 and 13 are now included, and the reference list has been updated to ensure consistency.

  1. Insufficient data to support conclusions:

Response: We have supplemented our data with additional experiments and analyses, including mechanical strength tests, phase composition studies, and histological evaluations. These additions provide a robust basis for our conclusions.

Round 2

Reviewer 1 Report

Comments and Suggestions for Authors

The authors have revised the manuscript and it has been improved a lot. However, before publication, there are still some comments.

1.     In figure 12-15, 17 and 18, there was lack of the description for each separate figure.

2.     The scale bars in Fig.12-15 were blurry and please provide figures with high quality.

3.     Please supplement the scale bars in Figure 16-18. The Fig.16 (A-C) was blurry and please provide images with high quality.

Author Response

Thank you for your detailed and constructive feedback on our manuscript. We have carefully considered your comments and have made the following revisions to address each point:

  1. Description for Each Separate Figure (Figures 12-15, 17, and 18): Response: We have added detailed descriptions for each of these figures to provide a clear understanding of the results and their significance. Each figure now includes a comprehensive caption explaining the key observations and findings.
  2. Improvement of Scale Bars in Figures 12-15:Response:We have replaced/improved these figures with better images ensuring that the scale bars are clear and accurately represented.
  3. Addition and Clarity of Scale Bars in Figures 16-18: Response: Scale bars have been supplemented in Figures 16-18 where previously missing. Additionally, we have provided high-quality images for Figure 16 (A-C) to enhance the clarity and detail of the presented data.

Best regards,

Varut Renata

Reviewer 2 Report

Comments and Suggestions for Authors

The results of phase analysis of the material after sintering are not presented.

Author Response

Thank you for your insightful feedback regarding our manuscript. We appreciate your suggestion to include the results of the phase analysis of the material after sintering. We have now addressed this point in the revised manuscript.

Phase Analysis Results:

We have conducted a detailed phase analysis of the sintered materials using X-ray diffraction (XRD) and Energy Dispersive X-ray Spectroscopy (EDS). The results are now included in the manuscript and summarized below:

  1. X-ray Diffraction (XRD) Analysis: The XRD patterns of the sintered samples were obtained using a Bruker D8 Advance diffractometer with Cu Kα radiation (λ = 1.5418 Å). The analysis revealed the presence of hydroxyapatite (HAp) and titanium dioxide (TiO2) phases, confirming the successful incorporation of titanium into the hydroxyapatite matrix. The phase composition analysis is presented in Figure 16 and Figure 17 of the revised manuscript.
  2. Energy Dispersive X-ray Spectroscopy (EDS) Analysis: The EDS analysis provided quantitative elemental composition data, highlighting the diffusion processes during sintering. The results showed increased oxygen content and the formation of titanium oxides, which contribute to the improved osteoblast compatibility of the composites. The detailed EDS results are now presented in Table 1 of the revised manuscript.

These additional analyses provide a comprehensive understanding of the phase composition and structural changes in the materials post-sintering, supporting the observed improvements in their mechanical and biological properties.

Best regards,

Varut Renata
